# Genetic and Epigenetic Variations of HPV52 in Cervical Precancer

**DOI:** 10.3390/ijms22126463

**Published:** 2021-06-16

**Authors:** Katharine J. Bee, Ana Gradissimo, Zigui Chen, Ariana Harari, Mark Schiffman, Tina Raine-Bennett, Philip E. Castle, Megan Clarke, Nicolas Wentzensen, Robert D. Burk

**Affiliations:** 1Department of Pediatrics, Albert Einstein College of Medicine, Bronx, NY 10461, USA; katharine.bee@dbv-technologies.com (K.J.B.); ana.gradissimo@einsteinmed.org (A.G.); zigui.chen@cuhk.edu.hk (Z.C.); ariana.harari@merck.com (A.H.); 2DBV Technologies, 92120 Montrouge, France; 3Department of Microbiology, The Chinese University of Hong Kong, Hong Kong, China; 4Division of Cancer Epidemiology and Genetics, National Cancer Institute, Bethesda, MD 20892, USA; schiffmm@exchange.nih.gov (M.S.); castlep@mail.nih.gov (P.E.C.); megan.clarke@nih.gov (M.C.); wentzenn@mail.nih.gov (N.W.); 5Division of Research, Kaiser Permanente Northern California, Oakland, CA 94612, USA; Tina.R.Raine-Bennett@kp.org; 6Department of Epidemiology & Population Health, Albert Einstein College of Medicine, Bronx, NY 10461, USA; 7Division of Cancer Prevention, National Cancer Institute, National Institutes of Health, Rockville, MD 20850, USA; 8Microbiology & Immunology, and Obstetrics, Gynecology & Women’s Health, Albert Einstein College of Medicine, Bronx, NY 10461, USA

**Keywords:** HPV52, papillomavirus, methylation, next generation sequencing, evolution, phylogeny, precancer

## Abstract

The goal of this study was to identify human papillomavirus (HPV) type 52 genetic and epigenetic changes associated with high-grade cervical precancer and cancer. Patients were selected from the HPV Persistence and Progression (PaP) cohort, a cervical cancer screening program at Kaiser Permanente Northern California (KPNC). We performed a nested case-control study of 89 HPV52-positive women, including 50 cases with predominantly cervical intraepithelial neoplasia grade 3 (CIN3) and 39 controls without evidence of abnormalities. We conducted methylation analyses using Illumina sequencing and viral whole genome Sanger sequencing. Of the 24 CpG sites examined, increased methylation at CpG site 5615 in HPV52 L1 region was the most significantly associated with CIN3, with a difference in median methylation of 17.9% (odds ratio (OR) = 4.8, 95% confidence interval (CI) = 1.9–11.8) and an area under the curve of 0.73 (AUC; 95% CI = 0.62–0.83). Complete genomic sequencing of HPV52 isolates revealed associations between SNPs present in sublineage C2 and a higher risk of CIN3, with ORs ranging from 2.8 to 3.3. This study identified genetic and epigenetic HPV52 variants associated with high risk for cervical precancer, improving the potential for early diagnosis of cervical neoplasia caused by HPV52.

## 1. Introduction

Cervical cancer is the fourth most common type of cancer affecting women worldwide and the fourth highest cause of death from cancer in women [1,2]. The highest disease burden occurs in low- and middle-income countries, where more than 90% of deaths occur due to poor access to screening and treatment [3]. In fact, in many areas of Africa, cervical cancer is the leading cause of cancer-related deaths [2]. Practically all cases of cervical cancer are caused by infection with the human papillomavirus (HPV). Of the over 200 types of HPV curated, at least 12 are considered to be high-risk (HR-HPV) [4]. Two of the high-risk types, HPV16 and HPV18, are responsible for roughly 70% of cases of cervical cancer, with HPV31, 33, 35, 45, 52, and 58 constituting the majority of the remaining cases [5,6]. The distribution of these HR-HPVs varies geographically across continents, particularly HPV52 and HPV58, whose prevalence is notably higher in cases of invasive cervical cancer in Asia [5,6], though also frequently detected in Brazil, according to a recent report [7]. For example, HPV52 is the fourth most frequent HPV type detected in cervical cancer cases in China [8]. Many countries in Africa and Asia have yet to introduce the recently FDA-approved nonavalent HPV vaccine that includes HPV52 [9], and until the vaccine is implemented, these regions will continue to encounter a high burden of HPV52-related cervix cancer deaths [10,11,12,13,14,15,16,17,18,19]. Given that HPV52 is a major circulating HPV type in various countries in Eastern Asia, where the cervical cancer incidence rates remain elevated, the genomic characterization of this HPV type is of epidemiological importance, with implications for cervical cancer screening strategies. In addition, HPV52 has also been detected in Bowen disease of the palm [20] and penile cancer [21], indicating an oncogenic potential at other epithelial sites.

Most HR-HPV types associated with cervical cancer are phylogenetically closely related, belonging to the species groups *Alphapapillomavirus* 9 (Alpha9, HPV16-related) or *Alphapapillomavirus* 7 (Alpha7, HPV18-related) [22]. HPV52 falls within the Alpha9 species group. HPV types are characterized by their nucleotide sequence similarity and differ by more than 10% in the L1 open reading frame by definition [23]. HPV types are further grouped into variant lineages and sublineages when the pairwise nucleotide sequences of their complete genomes differ by 1–10% or 0.5–1%, respectively [24].

The majority of HPV infections clear (i.e., become undetectable by commonly used HPV tests) naturally within the first two years after exposure to the virus [25]. However, approximately 5% of HR-HPV infections will progress to cervical intraepithelial neoplasia grade 3 (CIN3), the most specific histopathologic diagnosis of precancer [26]. Of these cases, up to one-third will eventually develop into invasive cervical cancer [27]. Although cytological screening (Pap test) has greatly reduced the disease burden in high-resource countries, it has a lower sensitivity than testing for HR-HPV. The high sensitivity as well as the strong negative predictive value of HR-HPV testing have led to changes in screening guidelines by including HPV primary screening [28]. The U.S. FDA has approved HPV tests for cervical cancer prevention that do not require cytology. As a result of screening and early detection, ablation or excision of precancerous lesions can prevent cervical cancer. However, these screening methods are still unable to differentiate between benign HR-HPV infections that will clear and those that will progress to cancer, making it difficult to effectively manage women who test positive for HR-HPV infection [29]. Therefore, identifying biomarkers that differentiate between more or less risky HR-HPV infections is highly desirable [30].

A potential biomarker of HR-HPV infections associated with cervix precancer/cancer is HPV DNA methylation [31]. Methylation is the covalent modification of the DNA base cytosine that occurs classically at CG dinucleotides in which the cytosine is 5′ to the guanine (CpG site). Previous studies of cases of HPV16 infection have shown that viral methylation levels at CpG sites in the L1 region were significantly higher in patients with CIN3 or CIN3+ (i.e., CIN3, adenocarcinoma in situ (AIS) or squamous cell cancer (SCC)) compared to those women with HPV16 infections with <CIN2 (for a review, see [31,32,33,34]). Similarly, highly elevated levels of viral methylation were seen in women with CIN3+ infected with HPV18, -31, or -45 when compared to women < CIN2 [35]. Therefore, increased levels of methylation associated with cervical precancer may serve as a predictive or diagnostic biomarker for risk of developing cervical cancer in HPV-positive women.

HR-HPVs have between 80 and 150 CpG sites across the genome, many of which are highly conserved across species types [31]. Increased methylation at two CpG sites in the L1 region were among those with the highest association between methylation levels and disease status; these two sites show very high topological conservation across the HR-HPVs belonging to the Alpha9 species (HPV16, -31, -33, -35, -52, and -58) [31,36]. The high levels of conservation as well as the strong correlation of methylation and disease status at these sites suggest a potential functional role for methylation in disease pathogenesis. Additionally, sequence variations among sublineages may have evolved through niche and host adaptation with higher oncogenic potential including changes that may create or eliminate CpG sites, affecting the potential for methylation of the viral genome.

We sought to expand our understanding of HPV52 at the genomic level. Therefore, we report on the levels of viral methylation in patients positive for infection with HPV52 at selective CpG sites across the L1, L2, E7, E1, and URR (upstream regulatory region) regions on the basis of conservation across related types. In addition, we sequenced the complete viral genome of HPV52-positive case and control samples to identify possible variants that may be more strongly associated with CIN3.

## 2. Results

### 2.1. CpG Methylation Determined by Pyrosequencing

HPV isolates from 89 women infected with HPV52 but no other HR types were selected from previously characterized cervical samples from the PaP cohort for analysis of viral methylation levels in patients with CIN3 and controls (<CIN2). Out of the 24 CpG sites analyzed across the genome, 22 CpG sites had equal or higher median levels of HPV52 methylation in women diagnosed with CIN3 as compared to control HPV52-positive women without histological or cytological abnormalities (Table 1).

The median methylation levels ranged from 1.2 to 35.9% for controls (*n* = 39) and from 1.2 to 51.2% for cases (*n* = 50). Overall, the L1 region showed the highest median levels of methylation compared to other analyzed regions of the viral genome in both cases and controls. The median differences in methylation for cases compared to controls ranged from −1.3 to 17.9% across all sites, and *p*-values ranged from 0.89 to 0.00026 (Table 1). The calculated odds ratios (ORs) ranged from 0.5 to 4.8. The CpG site 5615, located in L1, was the most significant site associated with CIN3, with a difference in median methylation of 17.9% (OR = 4.8; 95% confidence interval (CI) = 1.9–11.8) and had an area under the curve (AUC) of 0.73 (95% CI = 0.62–0.83). The majority of CpG sites with the most significant differences in median methylation levels between cases and controls clustered in the 5′ ends of both L2 and L1 regions.

Two single nucleotide polymorphisms were identified at sites 7111 and 7112 in the L1 region in six samples (C7111A and G7112A); both eliminated a CpG site. When these samples were removed from the statistical analyses, the difference in median methylation of cases compared to controls changed slightly, from 4.4% to 4.3%. No other single nucleotide polymorphisms (SNPs) were found that affected the presence of a CpG site among those sites that were analyzed for methylation.

### 2.2. CpG Methylation Determined by Next-Generation Sequencing (NGS)

In addition to pyrosequencing, methylation levels of bisulfite-converted viral DNA were also analyzed using NGS for 6 CpG sites in the L1 region that are highly conserved across multiple HR-HPVs [31,37]. For this analysis, 88 samples were analyzed; one sample was of insufficient quantity. The median levels of methylation ranged from 5.3 to 48.2% for controls and from 6.3 to 52.9% for cases (Table 2). A comparison of the pyrosequencing and NGS analyses revealed that the results from the two methods were correlated, with R squared values for each CpG site ranging from 0.571 to 0.773 (Appendix A). The CpG site at position 5615 had the largest difference between case and control samples, with median methylation values of 40.9% in controls and 46.8% in cases, with a OR of 3.67 (95% CI = 1.51–8.92) (Table 2), similar to methylation determined by pyrosequencing (see above).

### 2.3. Sanger Sequencing Analysis of Complete HPV52 Genomes

To examine possible associations of HPV52 variant lineages and SNPs with disease status and methylation levels, we obtained complete HPV52 genomic sequences for 83 of 89 samples, and partial sequences were obtained for four samples (Figure 1). The genomic sequence of an additional 23 HPV52 samples from a previous study (Qv) were added to the analyses for a total of 106 complete and 4 partial genomes [36]. A total of 781 nucleotide sites across the full 7942 bp genome (9.8%) were variable when compared to the HPV52 reference genome (data not shown). The most variable region was the URR with 13.7% overall nucleotide diversity, followed by the noncoding region 1 (NCR1) located between the E2 and E5 genes with 13.3% diversity. In addition to SNPs, various insertions and deletions were seen throughout the genome, the majority of which were located in the URR.

### 2.4. HPV52 Phylogenetic Analysis of 106 Isolates

On the basis of the alignment of the complete genomic sequence of 106 HPV52 isolates (the four isolates for which only partial sequence was obtained were omitted from these analyses), we assigned variant lineages and sublineages using a alphanumeric system, with the “A1” clade containing the reference genome as previously described [36]. The phylogenetic tree generated from the complete nucleotide sequences clustered HPV52 isolates into a total of five lineages, including nine sublineages, with the majority of isolates belonging to lineage A (*n* = 72) (Figure 2). One lineage, E, and two sublineages, B3 and C1, consisted of only one isolate each. The maximum nucleotide pairwise difference between the most dissimilar isolates was 2.2%, seen between isolates in the C and D lineages.

### 2.5. Association of Variant Lineage and Disease Status

To determine if there was an association between HPV52 SNPs and disease outcome, the OR values for each SNP position were calculated and plotted as a viral genome wide SNP analysis (VWAS) (Figure 3). The OR values appeared stratified, displaying horizontal clusters of SNPs with approximately equal ORs. Upon further analysis, these clusters constituted the SNPs diagnostic of a sublineage, consistent with previous findings [38,39,40]. Of all the sublineages, the C2 sublineage had the highest range of ORs (2.8–3.3) associated with CIN3 (Figure 3, see red rectangular box). Of the five samples in the C2 sublineage, four were diagnosed with CIN3 and one had normal cytology.

### 2.6. Analysis of Nucleotide Variations at CpG Sites

Of the 781 variable nucleotides found throughout the genome, 125 (16.0%) were located within a CpG site based on the HPV52 reference sequence. Of the 131 CpG sites in the reference sequence, 56 of these CpG sites were disrupted by a nucleotide change in one of the HPV52 isolates. An additional 69 SNPs resulted in the creation of a CpG site not found in the reference genome. Of the 125 SNPs that either created or destroyed CpG sites, 80 were found among all members of B2 sublineage.

## 3. Discussion

The results of the present study reflect the importance of understanding how both genetic and epigenetic variations of HR-HPVs are risk factors for the pathogenicity of a given infection. The association of cervical cancer and infection with HR-HPV has been well established, and it has become possible to distinguish increasingly refined levels of genomic diversity responsible for this association. For example, it was initially determined that HR-HPVs belong predominantly to the Alpha9 and Alpha7 clades. It is now possible to define the magnitude of association with cervical precancer/cancer of not only clade and species, but also lineages, sublineages, and even specific nucleotides [34,39,40,41]. An additional method of analyzing association of HR-HPV infection and cervical cancer is the use of methylation levels as a biomarker of disease severity. Results of this study add to the understanding of the oncogenic qualities of HPV52 associated with both genetic variation and epigenetic modification of the viral genome.

A primary aim of our study was to analyze the association of disease outcome with levels of viral methylation to further validate the use of this epigenetic modification as a biomarker for identifying at-risk individuals [31,34,37,41,42,43,44]. Our data indicate that multiple CpG sites across the HPV52 genome show differential methylation levels associated with CIN3 compared to HPV52-positive infections without high-grade disease, particularly within the L1 region. Results achieved through pyrosequencing were confirmed for several of the CpG sites by NGS, in accordance with previous results focused on CpG methylation determined by NGS [32,33]. Our results support previous findings in which elevated levels of viral methylation were associated with high-grade cervical neoplasia and cancer (for review, see [31,37,41,43,45]). Taken together, these data add to the growing body of knowledge that increased HPV methylation may serve as a biomarker to identify women with precancerous/cancerous lesions, particularly in the viral L1 region [46,47]. Moreover, HR-HPV DNA methylation has also been shown to be predictive in women infected with HIV (which harbor multiple HR-HPV-type infections), as it could be used to discriminate active HR-HPV infections while suggesting the “causal” HPV type responsible for the detected cervical abnormalities [48].

Another goal of this study was to analyze the relationship between HPV52 sequence variation and disease outcome. We identified a cluster of SNPs diagnostic of the HPV52 C2 sublineage that was associated with an increased risk of CIN3. Of interest was the fact that lineage E and sublineages B2/B3 and C2 contained the L83V amino acid substitution previously suggested to alter the ability of E6 to bind and degrade p53, inducing stronger antiapoptotic signals [49,50,51,52]. These findings are consistent with a higher pathogenicity of HPV52 sublineage C2. In line with our findings that HPV52 lineage C is associated with CIN3, an increased prevalence of cervical neoplasia and this lineage was also identified in a Taiwanese population [53], as well as in a study from Korea [52]. However, other studies that have analyzed associations of HPV52 variant lineages and cervical precancer/cancer have included few or no samples belonging to the C lineage [54,55,56,57,58,59]. Additionally, most similar studies have only examined sequence variations in small regions of the genome, most often fragments of E6, E7, L1, and URR, making it difficult to definitively assign lineage and sublineage classification to the viral isolate [52,53,54,55,56,57,60]. It is interesting to note that many of the SNPs diagnostic of the C2 sublineage clustered in the E7 and URR portions of the genome, suggesting a potential functional role for these regions that might confer greater pathogenicity in members of this sublineage. Moreover, the dominance of A1 sublineage in our study is consistent with the distribution of this particular sublineage in the Americas and Europe [7,53,58,61].

Although the mechanism by which viral methylation results in an increased risk of developing cervical precancer and cancer is currently unknown, a number of possibilities exist to explain the association. It is possible that HR-HPV CpG methylation influences gene expression and cell cycle disruption [62]. Alternatively, CpG methylation may be a marker of cell dedifferentiation associated with precancer and the changes in viral gene expression [63]. High levels of conservation seen at CpG sites across phylogenetically related HPV types as well as the correlation between viral methylation and cervical disease status suggest that, in fact, these sites may play an important functional role. Levels of methylation may be impacted by the number of CpG sites in the HPV genome, a factor that changes as a result of sequence variations between and within types that create and destroy CpG sites. Future studies should consider HPV sequence variation and its influence on number and topology of CpG sites and level of CpG methylation.

The current report has strengths and weaknesses. Some of the strengths are related to the population-based selection of cases and controls with HPV52 and that HPV52 was the only HR-HPV type present in the sample. Nevertheless, the study had limited number of cases and controls and did not include sufficient numbers of various population groups that might be more susceptible to HPV52 transformation. Thus, larger studies amongst other population groups are needed to validate the genomic and epigenetic contributions of HPV52-associated cervix cancer risks.

## 4. Materials and Methods

### 4.1. Study Design and Population

Patients were selected from the HPV Persistence and Progression (PaP) cohort, which include over 55,000 women who were enrolled in the cervical cancer screening program at Kaiser Permanente Northern California (KPNC). The cohort was created by banking residual, waste cervical specimens from women aged 25 and older who tested positive for hybrid capture 2 (HC2, Qiagen, Georgetown, MD, USA) during screening. Women with collected specimens were mailed opt-out consent letters and could respond if they wished not to have their specimens and information included in the study for HPV-related biomarkers for cervical precancer and cancer. Approximately 8% elected to opt-out and their specimens were destroyed. The study protocol was reviewed and approved by the Kaiser Foundation Research Institute and the National Cancer Institute (NCI) Institutional Review Boards. Patient cervical samples were screened using either conventional or liquid-based cytology and were tested for HPV by HC2, as previously described [64]. Samples positive for HC2 were analyzed for HPV type by PCR, as reported [64,65]. The current study consisted of a nested case–control population from the PaP cohort containing a total of 89 women 30 years of age or older that tested positive for HPV52 infection without other carcinogenic HR-HPV infections at the start of the study; 50 women were confirmed to have precancer by histology (49 CIN3 and 1 AIS) and 39 women were negative for intraepithelial lesions or malignancy (NILM) on cytologic exam and served as controls.

### 4.2. Sodium Bisulfite Treatment

DNA was treated with sodium bisulfite using the EpiTect Plus DNA Bisulfite Kit (Qiagen, Georgetown, MD, USA) as per the manufacturer’s protocol, which allows for the downstream conversion of unmethylated cytosine nucleotides to thymine and the preservation of methylated cytosines in the subsequent PCR reaction. Briefly, 15 μL of DNA was incubated with freshly prepared sodium bisulfite for approximately 5 h, cycling between 95 and 60 °C incubations, and held at 20 °C overnight. Following conversion, DNA samples were desulphonated and eluted in 25 μL of elution buffer. Bisulfite-converted DNA was stored at −20 °C or used immediately for PCR.

### 4.3. Pyrosequencing

Pyrosequencing assays were designed for the quantitation of methylated cytosines at 24 CpG sites across the HPV genome (located in viral genes E7, E1, L2, L1, and the URR) (see Table 1). Sites were chosen on the basis of a high level of conservation among HPV types belonging to the Alpha9 clade and therefore closely related to HPV52 [36]. Bisulfite-converted DNA was amplified by PCR using the PyroMark PCR kit (Qiagen, Georgetown, MD, USA). HPV52-specific primers were designed to amplify a ≈150 base pair (bp) region surrounding the CpG site (s). For each primer set, the reverse primer was 5′-biotinylated to permit downstream pyrosequencing of the PCR products. PCR amplification of HPV bisulfite-converted DNA was performed under the following conditions: initial denaturation at 95 °C for 15 min, 45 cycles of 94 °C 30 s, 55 °C 30 s, and 72 °C 30 s, and a final extension for 10 min at 72 °C. For some assays, the 55 °C annealing temperature was adjusted for optimization. PCR products were separated by electrophoresis on a 3% agarose gel. An aliquot of successful PCR products (a single band of anticipated size) was sent for pyrosequencing on a PSQ96 ID (Qiagen, Georgetown, MD, USA) at the Epigenomics Core Facility, Albert Einstein College of Medicine, Bronx, NY, USA.

### 4.4. Next-Generation Sequenicng (NGS)

DNA was bisulfite-converted and purified as described above. Methylation-sensitive PCR primers were designed using MethPrimer [66] and synthesized by IDT (Integrated DNA Technologies, Coralville, IA, USA), as previously reported [32]. A region of 153 bp containing a total of 6 CpG sites in L1 was amplified by PCR. For all samples, a unique 8 bp DNA Hamming barcode [67] flanked by a 3 bp (5′) and 4 bp (3′) padding sequence was introduced to both the 5′ and 3′ ends of the amplicon by the PCR primers. PCR products were analyzed by gel electrophoresis to confirm the presence of the predicted fragment size and to determine relative concentrations. Barcoded PCR products were pooled at approximately equal concentrations and purified by DNA electro-elution and isopropanol precipitation (QIAquick Gel Extraction Kit, Qiagen). Library preparation of the purified PCR products was performed using KAPA LTP Library Kit (Kapa Biosystems, Wilmington, MA, USA), and the pooled library was submitted for paired-end sequencing on an Illumina HiSeq2000 (Illumina Inc., San Diego, CA, USA) at the Epigenomics Core Facility, Albert Einstein College of Medicine, Bronx, NY, USA.

### 4.5. Bioinformatic Analysis of NGS Reads

Prior to analysis for methylation quantification, sequencing files were first demultiplexed on the basis of the unique barcodes using scripts generated *in-house* in order to obtain Illumina paired-end reads for each sample [32]. Reads were aligned with the HPV52 reference genome by Bowtie v0.12.9 [68], and the methylation status of each CpG site was determined by Bismark v0.7.7 [69] using the default quality score parameter set to Q30. The formula for determining the percent methylation was the ratio of the number of C reads divided by the number of C + T reads.

### 4.6. Sequencing of HPV52 Complete Genomes

The complete 8Kb genome of HPV52 was PCR-amplified in 2 to 3 overlapping fragments using type-specific primers with an equal mix of AmpliTaq Gold DNA polymerase (Applied Biosystems, Foster City, CA, USA) and Platinum Taq DNA Polymerase (Invitrogen, Carlsbad, CA, USA) as previously described [36]. Successful PCR products, as determined by gel electrophoresis to be of anticipated size, were isolated using the QuickStep 2 PCR Purification Kit (Edge BioSystems, Gaithersburg, MD, USA) or QIAquick Gel Extraction Kit (Qiagen, Georgetown, MD, USA) and submitted for Sanger sequencing at the Genomics Core Facility, Albert Einstein College of Medicine, Bronx, NY. Additional sequencing primers were applied to “walk” across each overlapping fragment. The complete genomes were assembled through mapping the obtained electropherogram to the HPV52 reference sequence (GenBank ID 397038).

### 4.7. Construction of Phylogentic Tree

The nucleotide sequences of the complete genomes were aligned to the reference sequence using the program MAFFT v6.846 [70]. Maximum likelihood (ML) trees were constructed using RAxML v7.2.8.27 [71] with optimized parameters on the basis of the aligned complete sequences linearized at the first “A” of the start codon of the E1 ORF. Bayesian trees were constructed using MrBayes v3.1.2 with 10,000,000 cycles for the Markov chain Monte Carlo (MCMC) algorithm [72]. Single-nucleotide polymorphisms (SNPs) within the HPV52 lineages/sublineages were identified from the global sequence alignments using MEGA5 v5 [73], which was used to calculate mean nucleotide differences and standard errors within and between sublineages. The p-distance method in MEGA5 v5 was used to calculate the genome–genome pairwise differences from the above alignments.

### 4.8. Statistical Analyses

For both pyrosequencing and NGS results, median methylation values for each CpG site were compared between cases and controls using the nonparametric Mann–Whitney *U* test. OR values were computed using methylation levels stratified at the highest tertile in controls, and the AUC estimates were calculated for individual CpG sites. Methylation values obtained by pyrosequencing and NGS were plotted and evaluated using a linear regression model. R squared values to evaluate the correlation between pyrosequencing and NGS for each CpG site were calculated in Microsoft Excel 2008.

Differences in nucleotides at SNP positions and methylation at CpG sites among cases and controls were examined. Logistic regression was performed on identified SNP positions comparing HPV52-positive subjects with CIN3 to the HPV52-positive control subjects (<CIN2). Only SNP positions with >2% variant frequency were analyzed. The reference group at each position was determined by setting the most prevalent nucleotide as the reference. The ORs and 95% CI were calculated, and results were plotted as OR values (see Figure 3). Differences in HPV52 lineage and CIN3 were examined using logistic regression models, comparing A lineage to B lineage and within a specific lineage (i.e., A1 vs. A2, A2 vs. B2, etc.). Other lineages were not included in these analyses due to small numbers of samples. All analyses were performed using SAS v9.3 (SAS Institute, Cary, NC, USA) with a *p*-value < 0.05 as statistically significant.

## Figures and Tables

**Figure 1 ijms-22-06463-f001:**
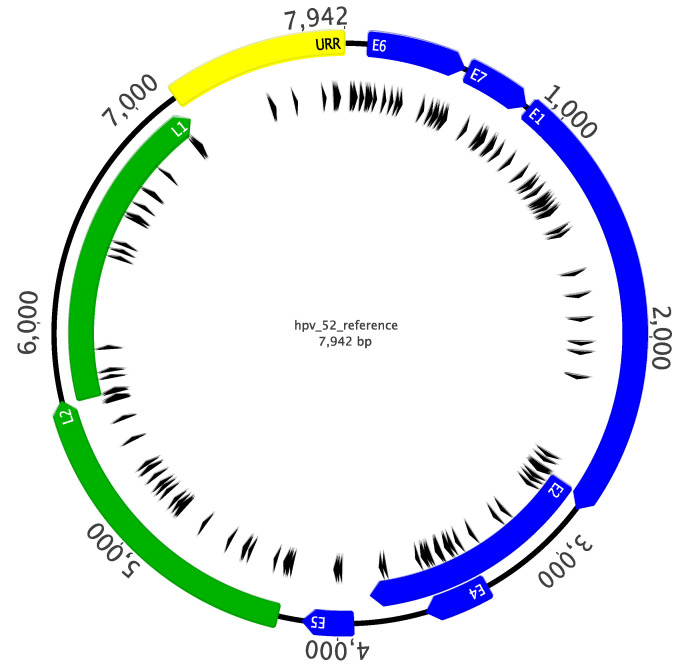
Distribution of CpG sites across HPV52 reference genome. The complete genomic sequence of the HPV52 genome (GenBank ID 397038) was extracted from the PaVE database (http://pave.niaid.nih.gov; accessed on 4 March 2021). CpG sites are marked with a black arrowhead. Early genes are shown in blue, late genes are shown in green, and the upstream regulatory region (URR) is shown in yellow. This image was created and annotated using Geneious v6.1.7 (http://geneious.com).

**Figure 2 ijms-22-06463-f002:**
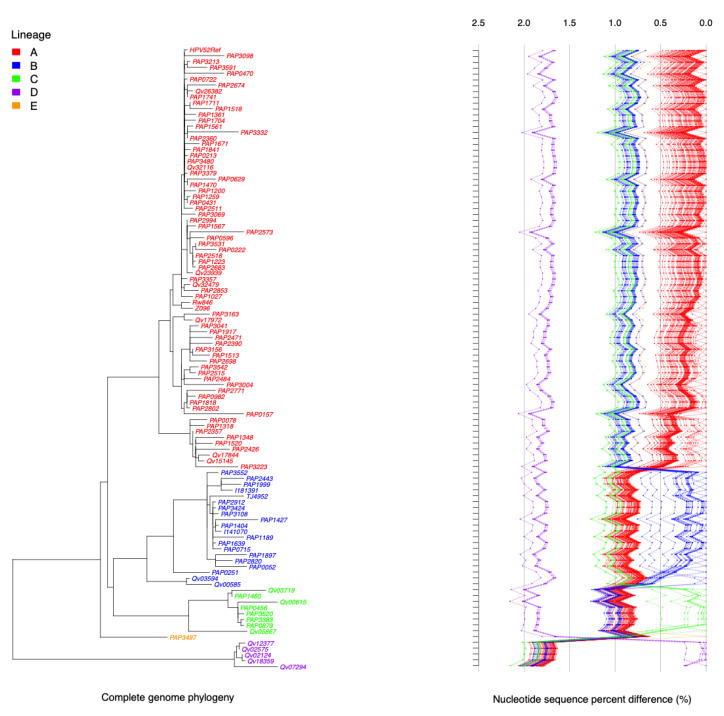
HPV52 phylogenetic tree topology and pairwise comparisons of individual complete genomes. The phylogenetic tree was inferred from global alignment of the complete nucleotide sequences from 106 HPV52 genomes. Distinct variant lineages (A, B, C, D, and E) were classified according to the topology and nucleotide sequence differences from >1% to <10% and are represented according to the color code shown at the top left of the figure. The percent nucleotide differences for each isolate compared to all other isolates are shown on the right. Lines indicate values for each comparison of a given isolate with the comparison to self as indicated by the 0% difference point to the far right.

**Figure 3 ijms-22-06463-f003:**
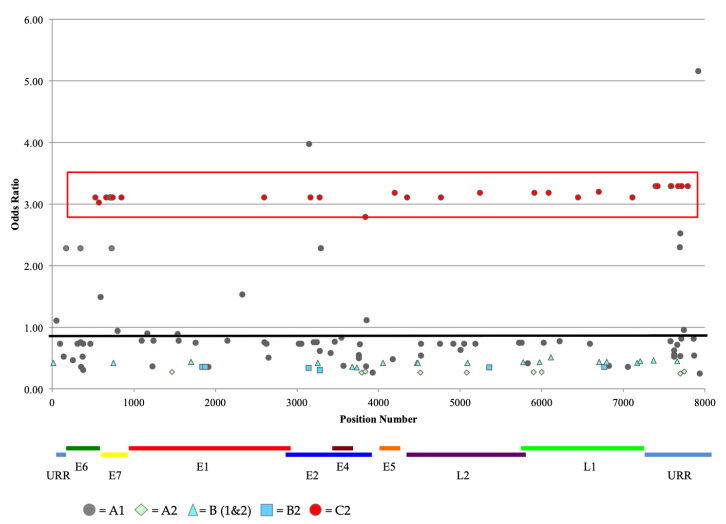
HPV52 viral genome-wide association study. Odds ratios for cases of CIN3 compared to HPV52 control infections were calculated for each SNP that occurred at least twice from the genomic sequence of 87 HPV52s. The red box represents variants associated with the C2 sublineage.

**Table 1 ijms-22-06463-t001:** Median methylation values of cases and controls by pyrosequencing.

		Median			Man-Whitney (Wilcoxon Rank Sum)	ROC Curve	95% CI Interval	Univariate Regression High HPV Methylation vs. Low/Medium Methylation		
Region Name	Genome Position	Control (*n* = 39)	Case (*n* = 50)	Difference	*p* Value	AUC		OR	95% CI	*p* Value
E7	c_754 c_776 c_779 c_787	1.40 13.20 1.20 5.90	1.80 13.30 1.20 6.40	0.40 0.10 0.00 0.50	0.010910 0.542481 0.771323 0.141489	0.6583 0.5379 0.5181 0.5916	(0.54, 0.78) (0.42, 0.66) (0.39, 0.65) (0.47, 0.71)	4.64 2.36 0.50 1.99	(1.88, 11.47) (0.95, 5.89) (0.18, 1.39) (0.82, 4.81)	0.000886 0.064880 0.182842 0.125934
E1	c_1122 c_1141	1.85 6.20	1.55 7.55	−0.30 1.35	0.319005 0.057360	0.4405 0.6187	(0.32, 0.56) (0.50, 0.74)	0.61 1.85	(0.23, 1.59) (0.76, 4.46)	0.313096 0.173033
L2	c_4274 c_4283 c_4292 c_4919 c_4926 c_4931 c_4949	8.00 18.50 10.90 2.40 4.30 4.40 32.60	13.50 17.20 19.90 3.00 5.25 4.70 39.50	5.50 −1.30 9.00 0.60 0.95 0.30 6.90	0.001171 0.003156 0.001699 0.085078 0.251942 0.295395 0.371808	0.7013 0.6831 0.6946 0.6062 0.5710 0.5649 0.5554	(0.59, 0.82) (0.57, 0.80) (0.58, 0.81) (0.49, 0.72) (0.45, 0.69) (0.45, 0.68) (0.43, 0.68)	2.44 2.25 2.44 2.25 1.50 1.56 1.38	(1.01, 5.86) (0.94, 5.41) (1.01, 5.86) (0.94, 5.41) (0.62, 3.63) (0.63, 3.84) (0.57, 3.35)	0.046613 0.070045 0.046613 0.070045 0.368990 0.333509 0.478098
L1	c_5606 c_5609 c_5615 c_5621 c_5648 c_5655 c_7098 c_7111 c_7119	18.00 15.00 30.20 5.00 26.80 35.90 3.10 5.90 3.40	33.20 25.90 48.10 8.05 32.15 51.20 3.90 10.30 5.70	15.20 10.90 17.90 3.05 5.35 15.30 0.80 4.40 2.30	0.001393 0.007199 0.000257 0.001651 0.034620 0.001799 0.024831 0.016845 0.032968	0.6982 0.6667 0.7267 0.6951 0.631 0.6936 0.6397 0.6489 0.6327	(0.58, 0.81) (0.55, 0.79) (0.62, 0.83) (0.58, 0.81) (0.51, 0.75) (0.58, 0.80) (0.52, 0.76) (0.53, 0.77) (0.51, 0.76)	3.38 2.64 4.78 2.86 2.08 3.67 1.99 2.76 3.55	(1.39, 8.17) (1.10, 6.36) (1.94, 11.80) (1.19, 6.90) (0.86, 5.00) (1.51, 8.92) (0.82, 4.81) (1.14, 6.68) (1.46, 8.65)	0.007037 0.030232 0.000684 0.019101 0.102620 0.004099 0.125934 0.024124 0.005258
URR	c_7557 c_7563	2.50 2.80	2.50 3.05	0.00 0.25	0.885024 0.423051	0.5096 0.553	(0.37, 0.64) (0.42, 0.68)	0.81 1.71	(0.30, 2.19) (0.68, 2.01)	0.680344 0.579475

Note: Pyrosequencing was conducted for cases (*n* = 50) and controls (*n* = 39) across 24 CpG loci, as described in the Materials and Methods section. Methylated loci are grouped to their location within the HPV52 genome.

**Table 2 ijms-22-06463-t002:** Median methylation values of cases and controls by next-generation sequencing.

		Median			Man-Whitney (Wilcoxon Rank Sum)	ROC Curve	95% CI Interval	Univariate Regression High HPV Methylation vs. Low/Medium Methylation		
Region Name	Genome Position	Control (*n* = 38)	Case (*n* = 50)	Difference	*p* Value	AUC		OR	95% CI	*p* Value
L1	c_5606 c_5609 c_5615 c_5621 c_5648 c_5655	27.36 18.84 40.90 5.34 32.21 48.22	30.45 22.11 46.76 6.26 37.07 52.92	3.09 3.27 5.86 0.92 4.86 4.70	0.005258 0.003606 0.003330 0.052489 0.003512 0.005895	0.6731 0.6805 0.6821 0.6203 0.681 0.6708	(0.56, 0.79) (0.56, 0.80) (0.57, 0.80) (0.50, 0.74) (0.57, 0.80) (0.55, 0.79)	3.38 1.92 3.67 1.92 2.86 2.86	(1.39, 8.17) (0.80, 4.61) (1.51, 8.92) (0.80, 4.61) (1.19, 6.90) (1.19, 6.90)	0.007037 0.146627 0.004099 0.146627 0.019101 0.019101

## Data Availability

The data presented in this study are openly available in GenBank, accession numbers: MZ374362-MZ374448.

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
