# Peer review of "Genetic and Epigenetic Variations of HPV52 in Cervical Precancer"

_ijms, 2021, doi:10.3390/ijms22126463_

Round 1
Reviewer 1 Report
Very well written article. The material and methods section is well described and sound and results are derived from a well described cohort using a nested cohort design. This work is complete and draws interesting conclusions in line with published work.
Very minor changes would be required.
1) On page 3, the authors state in the 2.1 paragraph that levels of HPV52 methylation in women with CIN3 were compared to levels in women wihtout lesion. However, the control group could include women with CIN1. Was this the case?
2) Considering figure 3, did some variations shared by different sublineages and if so, how was this taken into account?
3) Considering figure 3, to which sublineage belonged dits with ORs at ± 2.0
Reviewer 2 Report
The manuscript is solid and well designed, organized, and written.
Author Response
The authors wish to thank the positive feedback from the reviewer.